# Urine 5-Hydroxyindoleacetic Acid Negatively Correlates with Migraine Occurrence and Characteristics in the Interictal Phase of Episodic Migraine

**DOI:** 10.3390/ijms25105471

**Published:** 2024-05-17

**Authors:** Michal Fila, Jan Chojnacki, Marcin Derwich, Cezary Chojnacki, Elzbieta Pawlowska, Janusz Blasiak

**Affiliations:** 1Department of Developmental Neurology and Epileptology, Polish Mother’s Memorial Hospital Research Institute, 93-338 Lodz, Poland; michal.fila@iczmp.edu.pl; 2Department of Clinical Nutrition and Gastroenterological Diagnostics, Medical University of Lodz, 90-647 Lodz, Poland; jan.chojnacki@umed.lodz.pl (J.C.); cezary.chojnacki@umed.lodz.pl (C.C.); 3Department of Pediatric Dentistry, Medical University of Lodz, 92-217 Lodz, Poland; marcin.derwich@umed.lodz.pl (M.D.); elzbieta.pawlowska@umed.lodz.pl (E.P.); 4Faculty of Medicine, Collegium Medicum, Mazovian Academy in Plock, 09-402 Plock, Poland

**Keywords:** migraine, tryptophan metabolism, serotonin, 5-hydroxyindoleacetic acid, migraine-related anxiety

## Abstract

Although migraine belongs to the main causes of disability worldwide, the mechanisms of its pathogenesis are poorly known. As migraine diagnosis is based on the subjective assessment of symptoms, there is a need to establish objective auxiliary markers to support clinical diagnosis. Tryptophan (TRP) metabolism has been associated with the pathogenesis of neurological and psychiatric disorders. In the present work, we investigated an association between migraine and the urine concentration of TRP and its metabolites 5-hydroxyindoleacetic acid (5-HIAA), kynurenine (KYN), kynurenic acid (KYNA) and quinolinic acid (QA) in 21 low-frequency episodic migraine patients and 32 controls. We chose the interictal phase as the episodic migraine patients were recruited from the outpatient clinic and had monthly migraine days as low as 1–2 in many cases. Migraine patients displayed lower urinary levels of 5-HIAA (*p* < 0.01) and KYNA (*p* < 0.05), but KYN and QA were enhanced, as compared with the controls (*p* < 0.05 and 0.001, respectively). Consequently, the patients were characterized by different values of the 5-HIAA/TRP, KYN/TRP, KYNA/KYN, and KYNA/QA ratios (*p* < 0.001 for all). Furthermore, urinary concentration of 5-HIAA was negatively correlated with Migraine Disability Assessment score and monthly migraine and monthly headache days. There was a negative correlation between Patient Health Questionnaire 9 scores assessing depression. In conclusion, the urinary 5-HIAA level may be further explored to assess its suitability as an easy-to-determine marker of migraine.

## 1. Introduction

Migraine is the fifth (second in young women) cause of disability worldwide [1]. Despite the breakthrough in migraine therapy associated with the introduction of drugs targeting calcitonin gene-related peptide (CGRP) or its receptor and significant progress in neuroimaging, migraine diagnosis and therapy are not universal and have a high failure rate. This may result from poor knowledge of the mechanisms of migraine pathogenesis, which, in turn, may be a consequence of the restricted availability of the disease target tissue and limited value of animal models of human migraine as they reflect some aspects of the migraine syndrome and not the entire spectrum of symptoms [2]. Therefore, there is still a need for studies to expand current knowledge on the molecular mechanisms of migraine pathogenesis.

The diagnosis of migraine is based on information provided by patients that can be biased, despite the precise criteria of The International Classification of Headache Disorders ICH-3 [3]. The current ICHD-3 diagnostic criteria for migraine (migraine without aura, common migraine, hemicrania simplex) describe migraine as recurrent headache disorder manifesting in attacks lasting 4–72 h [3]. Typical characteristics of the headache are unilateral location, pulsating quality, moderate or severe intensity, aggravation by routine physical activity, and association with nausea and/or photophobia and phonophobia. The diagnostic criteria are at least five attacks lasting 4–72 h (when untreated or unsuccessfully treated) and headache has at least two of the following four characteristics: unilateral location, pulsating quality, moderate or severe pain intensity, aggravation by or causing avoidance of routine physical activity (e.g., walking or climbing stairs) and, during headache, at least one of the following: nausea and/or vomiting photophobia and phonophobia. Symptoms must not be better accounted for by another ICHD-3 diagnosis. There are several notes and comments for these criteria, including that one or a few migraine attacks may be difficult to distinguish from symptomatic migraine-like attacks. Therefore, sometimes clinicians may encounter problems with distinguishing migraine from migraine-like headaches. IHCD-3 also provides criteria for other kinds of migraine. Therefore, there is a need to supplement the traditional diagnostic tools with more objective molecular markers. This does not mean that these markers are to replace the ICHD-3-based diagnosis, but they may be useful in resolving doubts about the traditional diagnosis. Despite the difficulties in obtaining the target material, there are intense works on molecular markers of migraine (reviewed in [4]).

Migraine is a complex disease with many symptoms and many factors playing a role in its pathogenesis and these issues have been addressed in several excellent reviews, e.g., [5]. In general, the activation and sensitization of the trigeminal system may be crucial for migraine headache induction (reviewed in [6]).

Tryptophan (TRP) metabolism can be implicated in the pathogenesis of neurological and psychiatric disorders (reviewed in [7]). Tryptophan was reported to be associated with migraine in several studies, but it should be taken into account that they were performed in different types and phases of migraine (reviewed in [8]). A negative correlation between dietary intake of TRP and migraine risk was observed [9]. However, TRP is a precursor of several biologically active substances that may be responsible for the observed associations of migraine with TRP. 

After intake, the majority of TRP is metabolized in the digestive tract in the kynurenine (KYN), serotonin (5-HT), and indole pathways, which are competitively initiated by indoleamine 2,3-dioxygenase (IDO-1), TRP hydroxylase, and bacterial TRPase (TNA), respectively [10]. 5-Hydroxyindoleacetic acid (5-HIAA) is the main metabolite of serotonin produced by the action of monoamine oxidase and aldehyde dehydrogenase and excreted in the urine [11]. Consequently, 5-HIAA is employed to determine serotonin levels in the body. 

The KYN pathway of TRP metabolism gains an emerging role in migraine pathogenesis, which is supported by its involvement in the pathogenesis of functional gastrointestinal diseases and the functioning of the gut–brain–microbiota axis [8,12,13,14]. 

Although most of the ingested TRP is metabolized in the KYN pathway, 5-HT has an established role as a neurotransmitter and consequently is a natural candidate to play a role in migraine pathogenesis. This is supported by the identification of 5-HT as a blood vasoconstrictor, fitting the role of neurovascular incidence in the causative role of the trigeminal system in migraine pathogenesis [15]. Moreover, 5-HT receptors are widely distributed in the brain, including areas that are important in migraine [16]. The agonists of the 5-HT1 serotonin receptor have a long history of use in anti-migraine drugs [17]. Furthermore, serotonin is the precursor for the synthesis of melatonin in the pineal gland and melatonin may exert beneficial effects in migraine preventive and abortive treatment [18].

In this work, we investigated the urinary levels of TRP and its main metabolites: 5-HIAA, KYN, kynurenic acid (KYNA), and quinolinic acid (QA) in the 5-HT and KYN pathways in migraine patients and controls. Our working hypothesis was that the serotonin pathway of TRP metabolism evaluated by the urinary level of 5-HIAA might play a role in migraine pathogenesis. To explore this hypothesis, we associated some migraine characteristics related to the severity of the disease and its timing with the urinary concentration of 5-HIAA in migraine patients. 

Our primary motivation to choose the interictal phase was to explore some markers that may be attributed to migraine in general, and not to a particular headache. Secondly, we did not work with hospital patients who might donate urine at the moment determined in advance under professional control. Finally, we do not know how long it takes to express the consequences of the ictal phase of migraine into systemic changes and so the urine taken in the ictal phase might actually reflect features of the interictal phase. Moreover, the interictal phase is the most stable phase of migraine as it is sometimes difficult to distinguish between preictal, ictal, and postictal phases of the disease [19].

Migraine is frequently reported to be associated with mood disorders, symptoms of anxiety, and depression [20,21]. On the other hand, tryptophan metabolism may be implicated in mental health [22]. It was postulated that migraine patients with three or more headache days per month should be screened for anxiety symptoms [21]. Therefore, we looked for a correlation between the concentration of TRP metabolites and some indicators of anxiety and depression.

## 2. Results 

### 2.1. Characteristics of Migraine Patients

All subjects enrolled in this study displayed normal results of routine laboratory tests. 

All migraine patients suffered from low-frequency episodic migraine and eight of them experienced aura, mostly visual (Table 1). Non-steroidal anti-inflammatory drugs, triptans (sumatriptan, zolmitriptan, or almotriptan), and a gepant (rimegepant) were used as abortive drugs. Botulinum neurotoxin type A (BoNT/A) and monoclonal antibodies to calcitonin gene-related peptide (CGRP): erenumab, fremanezumab, and galkanezumab, were administrated in the preventive treatment. Most of the patients showed mild syndromes of mood disorders, anxiety, and depression. 

### 2.2. Tryptophan and Its Metabolites in Migraine Patients and Controls

Figure 1 presents the results of the determination of the urinary levels of TRP and products of its metabolism in migraine patients and controls. There were no differences between the urine levels of TRP in migraine patients and controls. Migraine patients had a lower level of 5-HIAA and KYNA than controls (*p* < 0.01 and *p* < 0.05, respectively), but they had higher levels of KYN and QA than controls (*p* < 0.05 and *p* < 0.001, respectively). The first interpretation of these results would be that migraine may be associated with a bias of tryptophan metabolism towards the kynurenine pathway, but such an immediate conclusion is not directly supported by a decrease in KYNA level. 

To explore further changes in TRP metabolism associated with migraine, some ratios of the concentrations of the metabolites were analyzed (Figure 2).

The 5-HIAA/TRP ratio was higher in migraine patients group than controls. That ratio indicates the activity of indoleamine 2,3-dioxygenase (IDO), playing a major role in the control of the kynurenine pathway of degradative TRP metabolism and, therefore, can be considered as a measure of the ratio of the TRP metabolism in the kynurenine pathway [23]. This was confirmed by an increased KYN/TRP ratio in migraine patients. The median of the KYN/KYNA ratio in the control individuals was about 6, but the median for migraine patients was lower by about half. These results suggest that KYNA was further metabolized in both groups.

### 2.3. Associations of Migraine Severity and Timing with Tryptophan Metabolites

We looked for a correlation between the clinical characteristics of migraine and parameters of TRP metabolism in the migraine patients group. First, we analyzed the correlation between the scores of the MIDAS (Migraine Disability Assessment Scale) questionnaire and the urinary levels of TRP, 5-HIAA, KYN, KYNA, and QA, as well as some of their ratios (Table 2).

The analysis of the MIDAS score with TRP, its metabolites, and their ratios showed a negative correlation between 5-HIAA and the 5-HIAA/TRP ratio.

To visualize an interindividual variability of the correlated parameters, we presented them along with a line adjusted by linear regression (Figure 3). Linear regression does not fit the data well, especially in the case of 5-HIAA. However, adjustments with nonlinear regression did not produce significantly better fits.

Next, we looked for a correlation between number of migraine days per month (MMDs) and tryptophan metabolism (Table 3). 

Similarly to MIDAS, MMDs were negatively correlated with the urinary concentration of 5-HIAA and the 5-HIAA/TRP ratio. To visualize these correlations, we presented the variability of the data along with the corresponding regression lines (Figure 4). 

Figure 4 shows a high interindividual variability of data and linear regression does not model the dependence between variables well. Moreover, there was not a functional relationship between MMDs and 5-HIAA or 5-HIAA/TRP ratio as some single values of the former corresponded to more than one value of the latter. This problem will be presented in the Discussion Section.

Next, we checked a correlation between the number of monthly headache days (MHDs) and TRP metabolism (Table 4). 

The number of headaches per month was correlated only with the urinary 5-HIAA concentration in migraine patients.

Figure 5 presents interindividual variability of MHD and urinary levels of 5-HIAA, along with a regression line. In general, the data displayed a decreasing tendency up to 10 MHD and an abrupt increase in urinary 5-HIAA concentration at 13 MHD may be occasional. The graphs in Figure 4 and Figure 5 may contain outliers, one per graph. These points can be identified as outliers with the ROUT (Prism) and Grubbs’ test with moderate aggressiveness. However, we did not remove these results as our population was too small to conclude the biological or medical significance of these results.

Other parameters characterizing migraine include the following: pain intensity evaluated by numeric rating scale (NRS), the number of headache attacks per month or quarter, duration of migraine, occurrence of aura, number of attacks per month, the use of gepants, triptans, onabotulinumtoxinA, and non-steroidal anti-inflammatory drugs in the abortive treatment and the use of anti-VEGF drugs, and other drugs in preventive treatment were not correlated with TRP metabolism. 

### 2.4. Association of Anxiety/Depression with Tryptophan Metabolites in Migraine Patients

As most of the migraine patients enrolled in this study showed mild symptoms of anxiety and depression, the next clinical characteristic we looked for a correlation with TRP metabolism was the Patient Health Questionnaire 9 (PHQ-9) score (Table 5). The PHQ-9 scale is a nine-item test that assesses the occurrence and severity of depression during the last two weeks before investigation [24].

We observed a negative correlation between PHQ-9 scores and urine concentration of 5-HIAA. We visualized the obtained results by a plot presenting the interindividual distribution of PHQ-9 scores and urinary 5-HIAA concentration (Figure 6). Moreover, two subgroups of migraine patients can be considered: one that has a strong negative correlation and another one that has a weak negative correlation between 5-HIAA levels and PHQ-9 scores.

That significant correlation between PHQ-9 and 5-HIAA prompted us to search for a correlation between Generalized Anxiety Disorder 7-item (GAD-7) scores and TRP metabolites, but we did not observe any significant association between them. To continue exploring the idea of the association of mood disorders/depression in migraine patients, we looked for a correlation between all characteristics of migraine we explored, i.e., MIDAS score, MMDs, MHD, duration of migraine, number of migraine headache attacks per month/quarter, PHQ-9 and GAD with vanillylmandelic acid (VMA), homovanillic acid (HVA) and xanthurenic acid (XA). All these substances can be related to tryptophan metabolism and brain functioning [25,26,27,28]. Moreover, it was shown that 5-HIAA correlated positively with HVA in cerebrospinal fluid of migraine patients [29]. However, calculations did not correlate significantly between migraine characteristics and VMA, HVA, or XA. 

Finally, we looked for a correlation between gastrointestinal symptom score (GIS) and TRP metabolism as we previously suggested that the kynurenine pathway of tryptophan metabolism may play a role in the pathogeneses of migraine and functional gastrointestinal disorders [13]. We did not obtain any correlation between GIS scale scores and tryptophan metabolism. 

## 3. Discussion

In the present work, we showed that patients with episodic migraine displayed different urinary levels of the TRP metabolites 5-HIAA, KYN, KYNA, and QA, as compared with the controls. Consequently, the patients were characterized by different values of the KYN/TRP, KYNA/KYN, and KYNA/QA ratios. No differences were observed for the urinary levels of TRP and the 5-HIAA/TRP ratio. On the other hand, urinary concentration of 5-HIAA was negatively correlated with MIDAS score, MMDs, MHD, and MIDAS score, and MMDs were also negatively correlated with the 5-HIAA/TRP ratio. Furthermore, a negative correlation of PHQ-9 with urinary 5-HIAA was observed.

The routine diagnosis of migraine is based on the reporting of symptoms by patients, with questions asked by a physician and the allocation of their answers on the scale(s) used by the physician. Therefore, such a diagnosis is subjective, so it is difficult to standardize the criteria for migraine diagnosis. Several indicators are postulated to be associated with migraine and so they can support the disease diagnosis. In general, migraine is considered a complex disease, with its pathogenesis underlined by the interaction of genetic and environmental factors. Genetic markers are proposed in monogenic migraines, including familial hemiplegic migraine and migraine with aura associated with hereditary small-vessel disorders [30]. Despite significant progress in genome analysis, the association of genetic factors with common migraine is still a big challenge [31]. Moreover, these markers are more useful in migraine therapy than diagnosis. At present, neuroimaging cannot be considered a routine method in common migraine diagnosis and is recommended in migraine with specific, rare symptoms (reviewed in [32]). Therefore, there is a need to establish migraine markers that could support the routine diagnosis of the disease. 

In our work, we pointed out a non-invasive, easy-to-determine parameter, 5-HIAA, which correlated with certain characteristics of migraine. Surely, it cannot be considered an independent marker of migraine, but its informative potential may be high as this TRP metabolite is excreted with urine. Our results suggest that 5-HIAA is strongly associated with migraine as it was correlated, either singly or in combination with TRP, with several migraine attributes, including MIDAS score, MMDs, MHD, and, additionally, with PHQ-9 scores.

Other studies associating migraine with 5-HIAA produced various results. However, we have not found any recent study on this subject. 5-HIAA is a representative of 5-hydroxyindoles (5-HIs), which were reported to fall during headache attacks in the blood of 17 out of 20 migraine patients [33]. However, another study performed on 14 migraine patients showed that urine concentrations of 5-HT and 5-HIAA fluctuated during and after migraine attacks [34]. The results of 5-HIAA variation were confirmed in a subsequent study performed in cerebrospinal fluid [35]. However, a 1976 study showed a significant increase in urine excretion of 5-HIAA during the early headache stage and a lack of correlation between some migraine characteristics and the levels of 5-HIAA in cerebrospinal fluids of migraine patients and a positive correlation with others [29]. A study with 9 migraine patients and 4 controls did not show any significant changes in the urine 5-HIAA in migraine patients relative to control individuals [35]. It was shown that migraine patients displayed lower plasma 5-HT and higher 5-HIAA levels than controls and patients with tension headaches [36]. However, during migraine attacks, plasma 5-HIAA concentrations were lower than in controls. A study performed on 8 migraine patients and 10 tension-type headache sufferers showed significantly decreased levels of 5-HIAA in the urine of both groups, as compared with controls [37]. Another study showed increased concentrations of serum 5-HIAA between attacks of migraine with aura and at the beginning of attacks of both migraine with and without aura [38]. The urinary level of 5-HIAA was observed not to change in young men but it decreased in female migraine patients when compared with their sex-matched controls [39]. No association was observed between 5-HIAA excretion and the characteristics of migraine. In summary, there are few studies investigating the urine level of 5-HIAA in migraine patients and the results of these studies are inconsistent; the most likely reasons for this inconsistency may be fluctuations in 5-HIAA levels during the disease course of different types of migraine and the weak statistical power of those studies. 

Migraine is frequently associated with symptoms of anxiety and/or depression (reviewed in [40,41]). This association is reported as bidirectional, i.e., migraine headaches may increase the risk of anxiety/depression and vice versa [42,43]. Therefore, there is a need to identify these migraine patients who require special psychiatric care. Our study showed that PHQ-9 scores were correlated with urinary 5-HIAA levels and, therefore, its determination may help to identify such patients. However, our results, based on a single self-reported questionnaire, cannot determine any role of 5-HIAA in migraine-associated mental disorders as it would require performing special psychological examinations by specialized personnel. However, several studies showed fluctuations in 5-HIAA levels in neurological and psychiatric disorders (reviewed in [44]). 

In our previous works, we considered tryptophan metabolism in the context of the gut–brain–microbiota axis [45,46,47,48]. However, those studies were performed on hospital patients, but this study enrolled patients of an outpatient clinic and most of them were not willing to donate blood or undergo the examination of the gut and the microbiota.

The two most important limitations of our study are the small number of enrolled individuals and the lack of standardization of TRP intake. The small number of individuals in our study resulted in a moderate statistical power of the tests we employed to analyze our data. However, we employed the resampling bootstrap technique to lower the chance of accepting a false hypothesis. We want to underline that many studies we cited in our discussion, published in highly impacted journals, enrolled even fewer patients than we did. Moreover, it could be considered that the determination of the sample size before research may not be the best solution, as it depends on the test that is to be used, which, in turn, depends on the distribution of data. We could not increase the number of patients due to limited financial and human resources, but the number of individuals in our cohort was in the typical range for 5′-HIAA/headache/migraine studies published so far. We did not interfere with the diet of enrolled individuals, but we did not observe any difference between the urine concentration of TRP in the patients and the control groups and the 5′-HIAA/TRP ratio was also the same in both groups. Another limitation of our study was to not measure the level of 3-hydroxykynurenine, a product of the KYN pathway of TRP metabolism, especially since this molecule is reported to exert a dual, pro-, or anti-oxidative action in the central nervous system [49]. Moreover, increased conversion of TRP to KYN with a decrease in KYNA/QA ratio suggests that the level of 3-hydroxykynurenine might change [50]. As stated above, our work did not include research on the dependence of the urinary 5-HIAA expression on other factors than migraine, including sex and age. Furthermore, 5-HIAA should be explored in different variants of migraine, e.g., migraine with aura, chronic migraine, especially in the context of the disease chronification, familial hemiplegic migraine, and migraine-like symptoms.

Although we did not observe changes in the KYN, KYNA, and QA levels, we cannot exclude alterations in the kynurenine pathway of TRP metabolism in our patients, as we did not measure the level of all metabolites of that pathway. In particular, 3-hydroxykynurenine levels were not measured. More evidence on the role of the KYN pathway of TRP metabolism in migraine and headaches can be found in other works, e.g., [12,13,51,52,53].

Urine 5-HIAA is employed as a proxy of serotonin and, consequently, it has a high clinical significance as a diagnostic and prognostic marker in many diseases [54]. The 5-HIAA urine test displays 100% specificity in carcinoid tumors [55]. It is difficult to unambiguously attribute the changes in 5-HIAA urinary level to a specific class of pathological condition as it was reported to increase and decrease in diseases classified into a single group, e.g., inborn errors of metabolism (reviewed in [11]). Also, in mental disorders, 5-HIAA was reported to change in opposite directions [45,56]. In our work, a lower level of urinary 5-HIAA was observed in episodic migraine patients. Still, in these patients, the 5-HIAA level was negatively correlated with PHQ-9 scores, which may be considered a measure of depression. The indoleamine hypothesis of depression says that the vulnerability to depression is underlined by the decreased serotonergic activity in the brain [57]. Such decreased activity may result in a lower level of 5-HIAA. This is in line with our results—our migraine patients had, on aveI have removed itrage, lower levels of 5-HIAA than controls, but migraine patients with high PHQ-9 scores, corresponding to depression, had lower levels of 5-HIAA than those with lower PHQ-9 scores. Therefore, our results agree with lower levels of 5-HIAA in migraine and depression reported in other studies [58]. However, our evidence on the association between low levels of 5-HIAA and depression is rather weak as the diagnosis of depression requires deeper analysis than a single self-reported test.

Although our study was conducted in the interictal phase of migraine, there is emerging evidence that symptoms associated with the headache phase may persist between migraine attacks [59]. These symptoms include allodynia, hypersensitivity, photophobia, phonophobia, osmophobia, visual/vestibular disturbances, and motion sickness. Therefore, the interictal phase of migraine should be further investigated to identify all migraine-related factors that may lower the quality of patients’ life. 

In summary, the urinary concentration of 5-HIAA is negatively correlated with migraine and its characteristics and may be further explored to assess its usefulness as a non-invasive and easy-to-determine molecular marker in migraine.

## 4. Materials and Methods

### 4.1. Patients and Ethics

A total of 21 patients with migraine and 32 age- and gender-matched migraine-free individuals (controls) were enrolled in this study and everyone signed informed consent to participate in this study. The participants were recruited from patients of the Department of Developmental Neurology and Epileptology, Polish Mother’s Memorial Hospital Research Institute in Lodz, and the Department of Neurology and Neurology Outpatient Clinic of the Regional County Hospital in Sieradz, Poland. The study was approved by the Bioethical Committee of Polish Mother Memorial Hospital Research Institute, Lodz, Poland, (permit no. 51/2020) and the Bioethical Committee of Medical University of Lodz, Lodz, Poland (permit no. RNN/176/18/KE).

### 4.2. Diagnosis

Migraine was diagnosed according to the criteria of the International Classification of Headache Disorders 3-beta (ICHD-3 beta) [3]. Only patients with episodic migraine cases were included in this study. The exclusion criteria for both groups included organic diseases of the central or peripheral nervous system, any form of functional neurological disorder, except for episodic migraine as per inclusion criteria for the patient group, advanced and/or decompensated, eventually exacerbated comorbidities, genetic diseases, mental disorders and diseases, congenital physical and/or intellectual disability, and a history of significant multiple traumas.

The control group was sex- and age-matched to migraine patients (Appendix A). The occurrence of headaches and signs of depression or/and anxiety, except for very occasional cases with recognized causes, was an exclusion criterion for the control group.

Mood disorder was assessed with a non-standard questionnaire based on the self-reported presence of 5 out of the 8 symptoms of sad mood, -hipo- or insomnia, feelings of guilt, decreased energy levels, decreased concentration, decreased appetite, decrease in pleasurable activities (anhedonia), and increased or decreased psychomotor activity [59]. Also, GAD-7, Generalized Anxiety Disorder 7-item, and PHQ-9, Patient Health Questionnaire 9, were used to assess patients’ mental health [60]. Original data for these analyses are contained in Appendix A.

The characteristics of migraine patients, including the disease duration, aura, frequency of attacks, treatment, and others were presented in Table 1. 

### 4.3. Determination of TRP Metabolites

All patients were informed to adhere to the dietary recommendations before donating urine samples, including refraining from eating fish and seafood 48 hrs and other tryptophan-rich foods 24 h before urine donation. Fasting urine samples for the analysis of TRP metabolites were collected into containers with a solution of 0.1% hydrochloric acid as a stabilizer. The urine samples were taken in the interictal period 10–30 days after the last migraine attack, which corresponded to the last administration of headache abortive drugs. L-tryptophan and its following metabolites: 5-HIAA, KYN, KYNA, and QA were determined in urine with liquid chromatography–tandem mass spectrometry (LC–MS/MS—Ganzimmun Diagnostics AG, Mainz, Germany; D-ML-13147–01-01), as described elsewhere [61] with some modifications [62]. The concentration of TRP and its metabolites were expressed in mg/g creatinine (mg/gCr). The concentration ratios 5-HIAA/TRP, KYN/TRP, KYNA/KYN, and KYNA/QA were calculated and presented in plots. Original data for these analyses are presented in Appendix A.

### 4.4. Data Analysis

The Shapiro–Wilk W test was used to check the normality of data distribution for each measured parameter ins the control and patient groups. If both controls and patients followed a normal distribution, the unpaired Student *t*-test was used to assess the significance of differences between groups; otherwise, the Mann–Whitney U test was used. The statistical power of the tests with a given sample size was calculated with G*Power v. 3.1 software [63]. Using that software, we calculated that the power of our studies ranged from 0.72 to 0.82 for a given sample size of 21 for migraine patients and 32 for controls with a significance level (alpha) of 0.05. To decrease the chance that the differences we observed resulted from pure chance, we employed the resampling bootstrap technique (10,000 iterations). The correlation between concentrations of TRP and its metabolites and characteristics of migraine were analyzed by Spearman’s rank test. All analyses were performed with STATISTICA 13.3 software (TIBCO Software INC., Palo Alto, CA, USA) and Resampling Stats Add-In for Excel v.4 (The Institute for Statistics Education, An Elder Research Company, Arlington, VA, USA). All original data for the correlation analyses are contained in Appendix A.

## Figures and Tables

**Figure 1 ijms-25-05471-f001:**
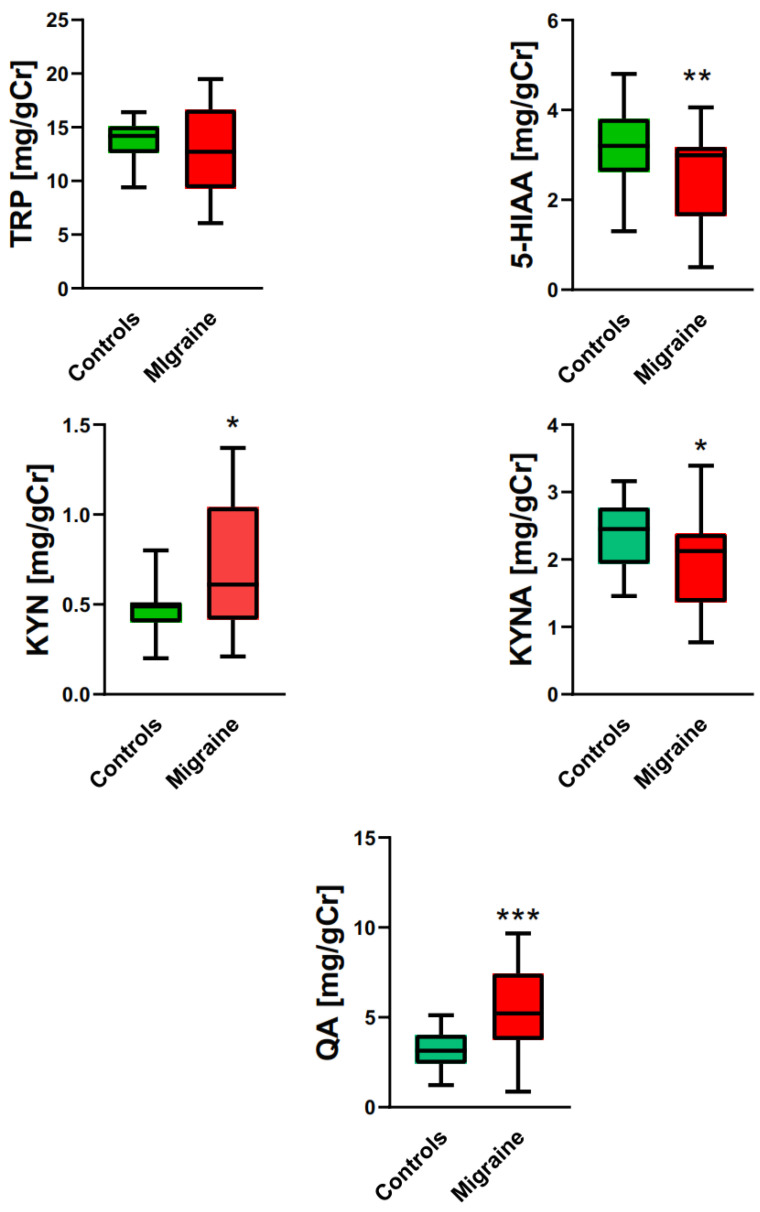
Tryptophan (TRP) and its main metabolites in migraine patients and controls. Urinary levels of tryptophan (TRP), 5-hydroxyaminoacetic acid (5-HIAA), kynurenine (KYN), kynurenic acid (KYNA), and quinolinic acid (QA) were expressed in milligrams per gram of creatinine (mg/gCr). The results are presented as median with boxes representing I and III quartiles and whiskers representing Min to Max values. Differences between migraine patients and controls were analyzed by Mann–Whitney U test; *n* = 21 in migraine patients group and *n* = 32 controls; *—*p* < 0.05; **—*p* < 0.01; ***—*p* < 0.001 as compared with controls.

**Figure 2 ijms-25-05471-f002:**
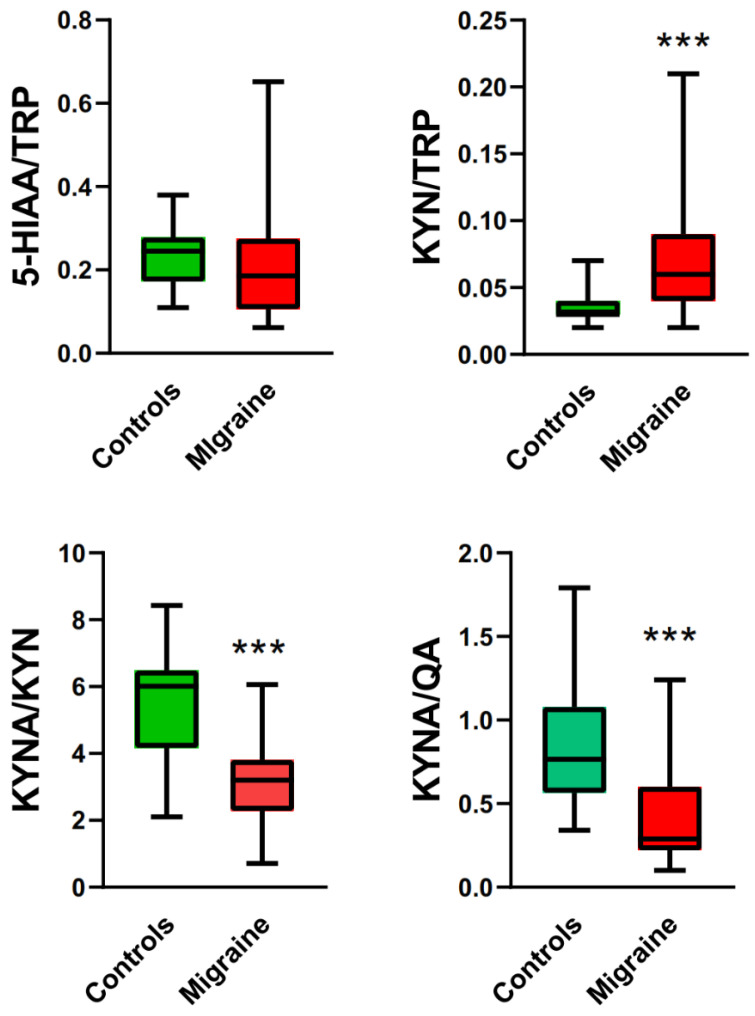
Ratios of urinary levels of 5-hydroxyaminoacetic acid (5-HIAA) and kynurenine (KYN) to tryptophan (TRP), KYN to kynurenic acid (KYNA), and KYNA to quinolinic acid (QA) in migraine patients and controls. Results are presented as the median with boxes representing I and III quartiles and whiskers representing Min to Max values. Differences between migraine patients and controls were analyzed by Mann–Whitney U test; *n* = 21 in migraine patients group and *n* = 32 in controls; ***—*p* < 0.001 as compared with controls.

**Figure 3 ijms-25-05471-f003:**
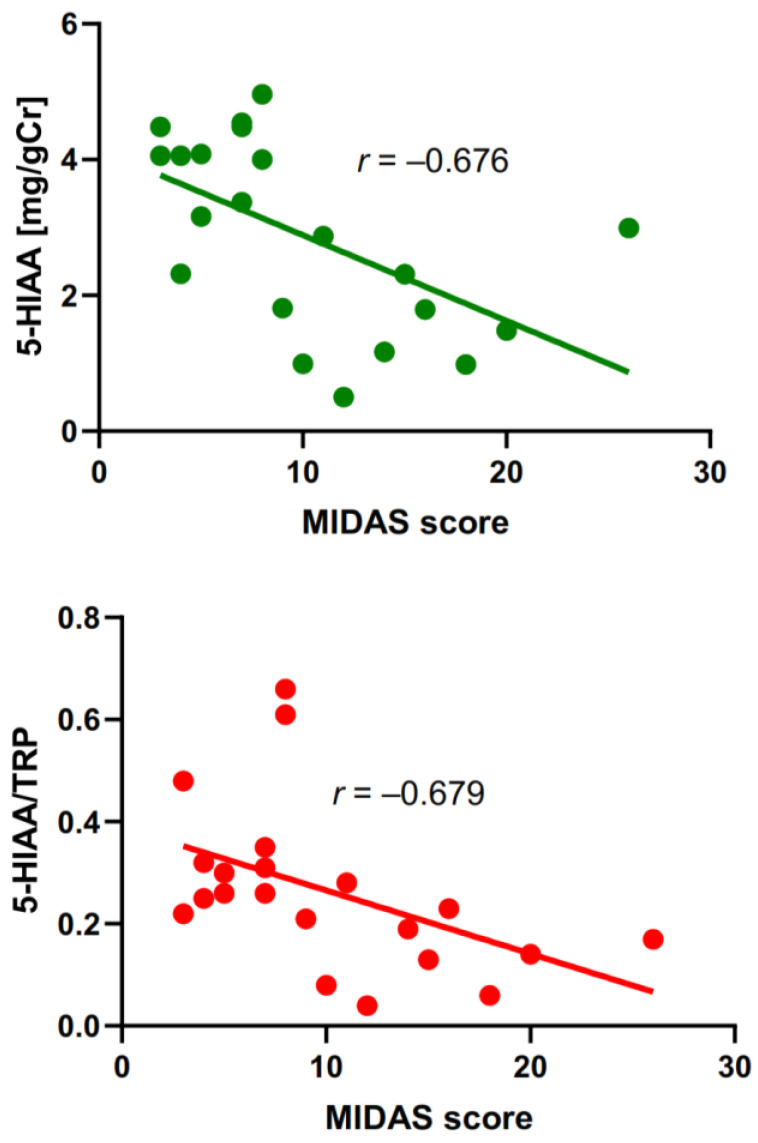
Correlation between the severity of migraine evaluated by Migraine Disability Assessment Scale (MIDAS) score and urine concentration of 5- hydroxyaminoacetic acid (5-HIAA) measured in mg per g of creatinine (gCr) (upper panel) or the ratio of 5-HIAA to tryptophan (TRP) (lower panel) in migraine patients. The correlation was assessed by the Spearman rank test with the rho rank coefficient (r). A linear regression line was drawn by the least square method with the equations Y = −0.126X + 4.150 (R^2^ = 0.312, upper panel) and Y = −0.012X + 0.390 (R^2^ = 0.226, lower panel).

**Figure 4 ijms-25-05471-f004:**
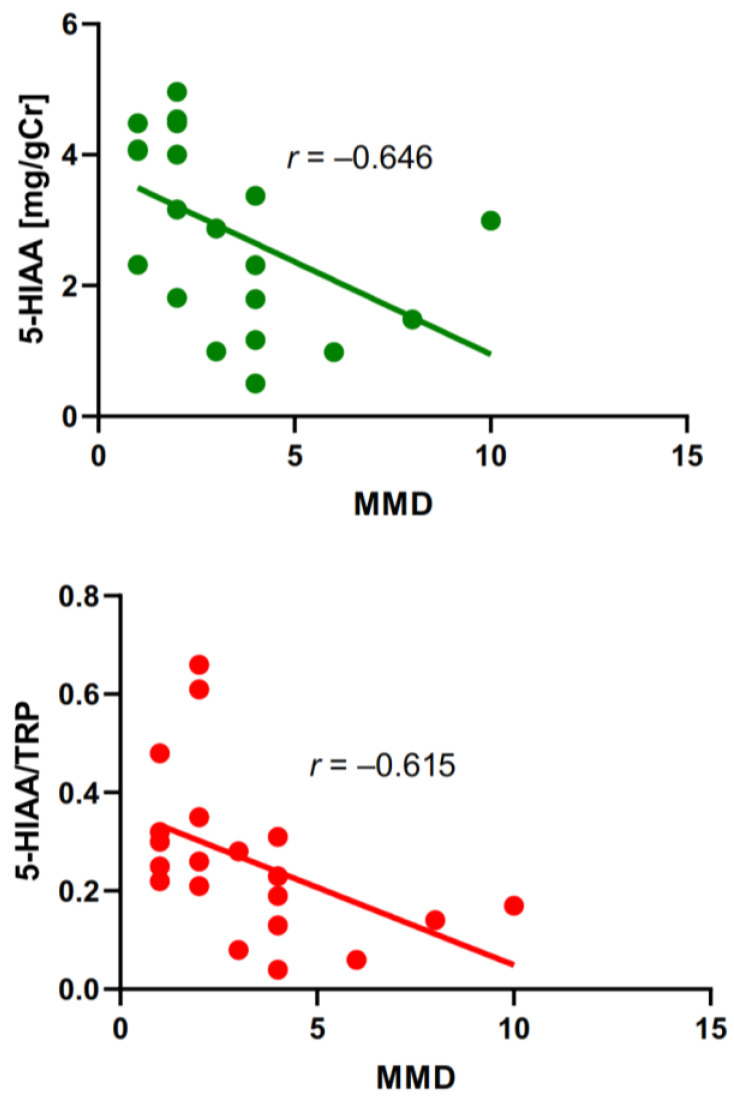
Correlation between monthly migraine days (MMDs) and urine concentration of 5- hydroxyaminoacetic acid (5-HIAA) measured in mg per g of creatinine (gCr) (upper panel) or the ratio of 5-HIAA to tryptophan (TRP) (lower panel) in migraine patients. The correlation was assessed by the Spearman rank test with the rho rank coefficient (r). The linear regression line was drawn by the least square method with the equations Y = −0.283X + 3.780 (R^2^ = 0.234, upper panel) and Y = −0.032X + 0.365 (R^2^ = 0.219, lower panel).

**Figure 5 ijms-25-05471-f005:**
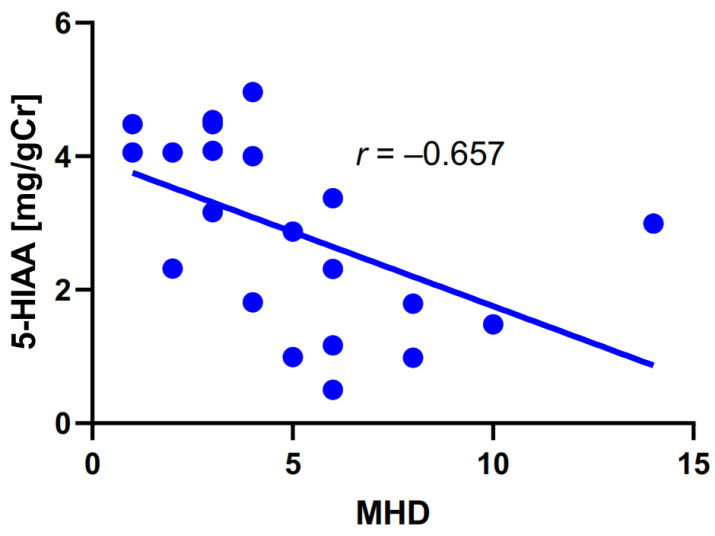
Correlation between monthly headache days (MHDs) and the urine concentration of 5- hydroxyaminoacetic acid (5-HIAA) measured in mg per g of creatinine (gCr) in migraine patients. The correlation was assessed by the Spearman rank test with the rho rank coefficient (r). The linear regression line was drawn by the least square method with the equation Y = −0.222X + 3.976 (R^2^ = 0.251).

**Figure 6 ijms-25-05471-f006:**
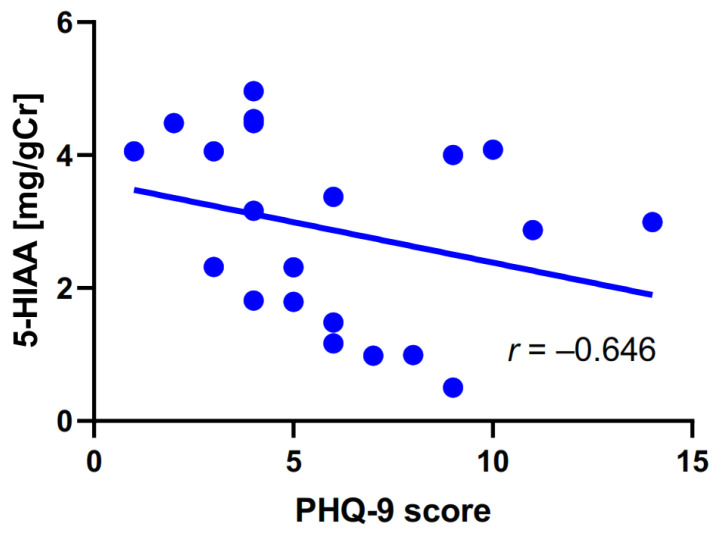
Correlation between the intensity of migraine headache evaluated by the Patient Health Questionnaire 9 (PHQ-9) and urine concentration of 5-hydroxyaminoacetic acid (5-HIAA) measured in mg per g of creatinine (gCr) in migraine patients. The correlation was assessed by the Spearman rank test with the rho rank coefficient (r). The linear regression line was drawn by the least square method with the equation Y = −0.122X + 3.601 (R^2^ = 0.080).

**Table 1 ijms-25-05471-t001:** Characteristics of migraine patients enrolled in this study (*n* = 21).

Characteristic	Specification (Mean ± SD (Range) or Type and Number)
Age	39 ± 13 (26–65)
Sex	14 F, 7 M
Migraine type	Episodic, 21
Aura	8
Frequency of attacks (per month)	1.2 ± 1.1 (0.3–4.0)
Frequency of attacks (per quarter)	3.5 ± 2.9 (1–12)
Time since diagnosis (years)	18 ± 12 (3–40)
Treatment	Abortive 16, prophylactic 5, both 3
Pain intensity (NRS ^1^)	7.0 ± 1.5 (5–10)
MIDAS	10.0 ± 5.9 (3–26)
Monthly migraine days	3.2 ± 2.4 (1–10)
Monthly headache days	5.0 ± 3.1 (1–14)
Mood disorders	8
Anxiety (GAD-7 score)	4.4 ± 3.1 (0–12)
Depression (PHQ-9 score)	6.0 ± 3.2 (1–14)

^1^ NRS, Numeric Rating Scale; MIDAS, Migraine Disability Assessment; GAD-7, Generalized Anxiety Disorder 7-item; PHQ-9, Patient Health Questionnaire 9.

**Table 2 ijms-25-05471-t002:** Correlation between the intensity of migraine headache evaluated by the MIDAS ^(1)^ questionnaire scores and the urinary levels of tryptophan (TRP), 5-hydroxyaminoacetic acid (5-HIAA), kynurenine (KYN), KYNA, and quinolinic acid QA, as well as their ratios in migraine patients. The correlations were analyzed with the Spearman rank test with the rho rank correlation coefficient.

MIDAS Score and	rho–Spearman	*p*
TRP	0.163	0.309
**5-HIAA**	**−0.676**	**0.001**
KYN	0.242	0.304
KYNA	−0.050	0.835
QA	0.145	0.543
**5-HIAA/TRP**	**−0.679**	**0.001**
KYN/TRP	−0.184	0.437
KYNA/KYN	0.077	0.748
KYNA/QA	−0.158	0.506

^(1)^ MIDAS, Migraine Disability Assessment Scale.

**Table 3 ijms-25-05471-t003:** Correlation between monthly migraine days (MMDs ^(1)^) and the urinary levels of tryptophan (TRP), 5-hydroxyaminoacetic acid (5-HIAA), kynurenine (KYN), KYNA, and quinolinic acid QA, as well as their ratios in migraine patients. The correlations were analyzed with the Spearman rank test with the rho rank correlation coefficient.

MMDs and	rho–Spearman	*p*
TRP	0.195	0.410
**5-HIAA**	**−0.646**	**0.002**
KYN	−0.188	0.427
KYNA	0.002	0.995
QA	0.143	0.548
**5-HIAA/TRP**	**−0.615**	**0.003**
KYN/TRP	−0.116	0.627
KYNA/KYN	0.042	0.862
KYNA/QA	−0.108	0.651

^(1)^ MMDs—monthly migraine days; TRP—tryptophan, 5-HIAA—5-hydroxyaminoacetic acid; KYN—kynurenine KYNA—kynurenic acid; QA—quinolinic acid.

**Table 4 ijms-25-05471-t004:** Correlation between monthly headache days (MHDs) and the urinary levels of tryptophan (TRP), 5-hydroxyaminoacetic acid (5-HIAA), kynurenine (KYN), KYNA, and quinolinic acid (QA), as well as their ratios in migraine patients. The correlations were analyzed with the Spearman rank test with the rho rank correlation coefficient.

MHD and	rho–Spearman	*p*
TRP	0.149	0.518
**5-HIAA**	**−0.657**	**0.001**
KYN	−0.206	0.369
KYNA	−0.051	0.826
QA	0.159	0.491
5-HIAA/TRP	−0.418	0.060
KYN/TRP	−0.152	0.510
KYNA/KYN	0.032	0.890
KYNA/QA	−0.136	0.556

**Table 5 ijms-25-05471-t005:** Correlation between the intensity of migraine headache evaluated by the PHQ-9 ^(1)^ questionnaire and the urinary levels of tryptophan (TRP), 5-hydroxyaminoacetic acid (5-HIAA), kynurenine (KYN), KYNA, and quinolinic acid (QA), as well as their ratios in migraine patients. The correlations were analyzed with the Spearman rank test with the rho rank correlation coefficient.

PHQ-9 Score and	rho–Spearman	*p*
TRP	0.004	0.987
**5-HIAA**	**−0.427**	**0.048**
KYN	−0.334	0.139
KYNA	−0.160	0.489
QA	0.216	0.347
5-HIAA/TRP	−0.307	0.176
KYN/TRP	0.042	0.862
KYNA/KYN	0.238	0.299
KYNA/QA	−0.198	0.390

^(1)^ PHQ-9, Patient Health Questionnaire 9.

## Data Availability

Anonymized data may be obtained from the corresponding author upon a reasonable request.

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
