# Peer review of "Urine 5-Hydroxyindoleacetic Acid Negatively Correlates with Migraine Occurrence and Characteristics in the Interictal Phase of Episodic Migraine"

_ijms, 2024, doi:10.3390/ijms25105471_

Round 1

Reviewer 1 Report (New Reviewer)

Comments and Suggestions for Authors

this is a very intersting topic but it can be improved, please add a table qith the result variables in both groups and the p values of the tests of difference between groups

Author Response

Comment: his is a very intersting topic but it can be improved, please add a table qith the result variables in both groups and the p values of the tests of difference between groups

Answer: We cannot repeat results presented in graphs as numeric values in tables. All the details can be read from the graphs and the p values are in the figure legends.

Reviewer 2 Report (New Reviewer)

Comments and Suggestions for Authors

The manuscript “Urine 5-Hydroxyindoleacetic Acid Negatively Correlates with Migraine Occurrence and Characteristics in the Interictal Phase of Episodic Migraine” presented by Fila et al. describes the results of work dedicated to finding clinically acceptable biomarkers of migraine. Levels of several TRP metabolism-associated compounds were determined in the urine of migraine-affected and control groups of patients. Levels of 5-HIAA were found among several other factors as important candidates for migraine diagnosis and assessment of its severity allowing to use biochemical parameters instead of psychological assessments in clinical practices. The work is well-planned, executed, and described. The main drawback is the absence of actual data as supplemental materials. These data are needed for proper manuscript review and future meta-analysis by other groups.

Additional comments (Comments are made during continuous reading of the manuscript. Therefore, answers to some raised questions may occur later in the text.)

Abstract

1) The abstract does not specify why the interictal phase was chosen for investigation and how it relates to migraine diagnostics. Add more description on the phase and its choice for this study in the Introduction as well.

2) What results are statistically significant? Such a statement is important for quick evaluation of the results.

Introduction

1) The following should be more nuanced “it is undiagnosed and untreated”. There are some approaches for the diagnosis and treatment of migraine. However, they are not universal and have a high failure rate. 

2) I would not use the word “inconsistent” in the following sentence “their results are inconsistent as they were performed in different types and phases of migraine (reviewed in [6]).”. Probably, they are consistent when a detailed migraine mechanism is understood. 

3) It is worth mentioning in which situations 5-HIAA urine levels are above and below normal. For example, “Lower levels have correlated with aggressive or violent behavior, depression, and obsessive-compulsive disorder.[21]” https://www.ncbi.nlm.nih.gov/books/NBK551684/ . This also can be accompanied in the Introduction or Discussion sections by a description of how the current study is different from the evaluation of depression using the same biomarker. Depression is one of the symptoms of migraine. 

There is a subsection in Results describing the correlation between 5-HIAA levels and PHQ-9.  Despite a small number of patients involved, it is worth, at least in the form of hypothesizing, to mention that there are two groups of two groups – one which has a strong negative correlation and another one which has a weak negative correlation of the parameters. 

The Discussion section has a paragraph dedicated to this topic and starts with the sentence “Migraine is frequently associated with symptoms of anxiety and/or depression (reviewed in [39,40]).”. However, it is too general to understand how similar the presented results and data on 5-HIAA levels in depression are. This comparison is critical for understanding how specific can be 5-HIAA as a migraine biomarker. 

Results 

1) “2.1. Characteristics of Migraine Patients” section should have information on the control group as well as a demonstration of no statistical difference between the groups in comparable parameters e.g. age, sex etc.

2) Figures 3, 4, and 6 have legends misplaced.

3) There is one likely outlier in MHD and MMD data sets. Was the test for outliers performed? 

4) Fig 3-6 have to have results of control measurements presented as well. Visualization of variability in the control data set is important for understanding of the results. It can be a data point distribution at 0 of X-axes or a separate panel.

Comments on the Quality of English Language

OK

Author Response

Comment: The manuscript “Urine 5-Hydroxyindoleacetic Acid Negatively Correlates with Migraine Occurrence and Characteristics in the Interictal Phase of Episodic Migraine” presented by Fila et al. describes the results of work dedicated to finding clinically acceptable biomarkers of migraine. Levels of several TRP metabolism-associated compounds were determined in the urine of migraine-affected and control groups of patients. Levels of 5-HIAA were found among several other factors as important candidates for migraine diagnosis and assessment of its severity allowing to use biochemical parameters instead of psychological assessments in clinical practices. The work is well-planned, executed, and described. The main drawback is the absence of actual data as supplemental materials. These data are needed for proper manuscript review and future meta-analysis by other groups.

Answer: We have added actual data: raw and transformed along with some details of statistical analysis as Supplementary Materials.

Additional comments (Comments are made during continuous reading of the manuscript. Therefore, answers to some raised questions may occur later in the text.)

Abstract

Comment: 1) The abstract does not specify why the interictal phase was chosen for investigation and how it relates to migraine diagnostics. Add more description on the phase and its choice for this study in the Introduction as well.

Answer: We have added the following sentence in the abstract (we cannot write too much as there is a word limit and the abstract of an experimental work should first present the obtained results):

“We chose the interictal phase as our episodic migraine patients were recruited from the outpatient clinic with diagnosed monthly migraine days as low as 1-2 in many cases.”

We have added the following fragment to the introductory section:

“Our primary motivation to choose the interictal phase was to explore some markers that may be attributed to migraine in general, and not to a particular headache. Secondly, we did not work with hospital patients who might donate urine at the moment determined in advance under professional control. Finally, we do not know how long it takes to express the consequences of the ictal phase of migraine into systemic changes and so the urine taken in the ictal phase might actually reflect features of the interictal phase. Moreover, the interictal phase is the most stable phase of migraine as it is sometimes difficult to distinguish between preictal, ictal and postictal phases of the disease [17]. “

with the new reference:

  1. Peng, K.P.; May, A. Redefining migraine phases - a suggestion based on clinical, physiological, and functional imaging evidence. Cephalalgia: an international journal of headache 2020, 40, 866-870, doi:10.1177/0333102419898868.

Comment: 2) What results are statistically significant? Such a statement is important for quick evaluation of the results.

Answer: Although in our opinion saying “greater”, “lower”, “different” etc. in science means “statistically significantly greater”, “statistically significantly lower”, and “statistically significantly different”, respectively, in response to this comment we have added the p values to each cited result.

Introduction

Comment: 1) The following should be more nuanced “it is undiagnosed and untreated”. There are some approaches for the diagnosis and treatment of migraine. However, they are not universal and have a high failure rate.

Answer: We have changed the sentence:

“Migraine is the fifth (second in young women) cause of disability worldwide, but despite its high prevalence, it is undiagnosed and untreated [1].”

into the following fragment:

“Migraine is the fifth (second in young women) cause of disability worldwide [1]. Despite the breakthrough in migraine therapy associated with the introduction of drugs targeting calcitonin gene-related peptide (CGRP) or its receptor and significant progress in neuroimaging, migraine diagnosis, and therapy are not universal and have a high failure rate.”

Comment: 2) I would not use the word “inconsistent” in the following sentence “their results are inconsistent as they were performed in different types and phases of migraine (reviewed in [6]).”. Probably, they are consistent when a detailed migraine mechanism is understood.

Answer: We have changed the sentence:

“Tryptophan was reported to be associated with migraine in several studies, but their results are inconsistent as they were performed in different types and phases of migraine (reviewed in [6]).“

into:

“Tryptophan was reported to be associated with migraine in several studies, but it should be taken into account that they were performed in different types and phases of migraine (reviewed in [6]).”

Comment: 3) It is worth mentioning in which situations 5-HIAA urine levels are above and below normal. For example, “Lower levels have correlated with aggressive or violent behavior, depression, and obsessive-compulsive disorder.[21]” https://www.ncbi.nlm.nih.gov/books/NBK551684/ . This also can be accompanied in the Introduction or Discussion sections by a description of how the current study is different from the evaluation of depression using the same biomarker. Depression is one of the symptoms of migraine.

Answer: We have added the following fragment to the Discussion section:

“Urine 5-HIAA is employed as a proxy of serotonin and consequently it has a high clinical significance as a diagnostic and prognostic marker in many diseases [56]. The 5-HIAA urine test displays 100% specificity in carcinoid tumors [57]. It is difficult to unambiguously attribute the changes in 5-HIAA urinary level to a specific class of pathological condition as it was reported to both increase and decrease in diseases that are classified into a single group, e.g., inborn errors of metabolism (reviewed in [58]). Also, in mental disorders, 5-HIAA was reported to change in opposite directions [59,60]). In our work, a lower level of urinary 5-HIAA was observed in episodic migraine patients. Still, in these patients, the 5-HIAA level was negatively correlated with PHQ-9 scores that may be considered a measure of depression. The indoleamine hypothesis of depression says that the vulnerability to depression is underlined by the decreased serotonergic activity in the brain [61]. Such decreased activity may result in a lower level of 5-HIAA. This is in line with our results – our migraine patients had on average lower levels of 5-HIAA than control, but migraine patients with high PHQ-9 scores, corresponding to depression, had lower levels of 5-HIAA than those with lower PHQ-9 scores. Therefore, our results agree with lower levels of 5-HIAA in migraine and depression reported in other studies [62]. However, our evidence on the association between low levels of 5-HIAA and depression is rather weak as the diagnosis of depression requires more deep analysis than a single self-reported test.”

with new references:

  1. Adaway, J.E.; Dobson, R.; Walsh, J.; Cuthbertson, D.J.; Monaghan, P.J.; Trainer, P.J.; Valle, J.W.; Keevil, B.G. Serum and plasma 5-hydroxyindoleacetic acid as an alternative to 24-h urine 5-hydroxyindoleacetic acid measurement. Ann Clin Biochem 2016, 53, 554-560, doi:10.1177/0004563215613109.
  2. Ito, T.; Lee, L.; Jensen, R.T. Carcinoid-syndrome: recent advances, current status and controversies. Curr Opin Endocrinol Diabetes Obes 2018, 25, 22-35, doi:10.1097/med.0000000000000376.
  3. Lenchner, J.R.; Santos, C. Biochemistry, 5 Hydroxyindoleacetic Acid. In StatPearls; StatPearls Publishing Copyright © 2024, StatPearls Publishing LLC.: Treasure Island (FL), 2024.
  4. Jayamohananan, H.; Manoj Kumar, M.K.; T, P.A. 5-HIAA as a Potential Biological Marker for Neurological and Psychiatric Disorders. Adv Pharm Bull 2019, 9, 374-381, doi:10.15171/apb.2019.044.
  5. Spreux-Varoquaux, O.; Alvarez, J.C.; Berlin, I.; Batista, G.; Despierre, P.G.; Gilton, A.; Cremniter, D. Differential abnormalities in plasma 5-HIAA and platelet serotonin concentrations in violent suicide attempters: relationships with impulsivity and depression. Life sciences 2001, 69, 647-657, doi:10.1016/s0024-3205(01)01158-4.
  6. Maes, M.; Leonard, B.E.; Myint, A.M.; Kubera, M.; Verkerk, R. The new '5-HT' hypothesis of depression: cell-mediated immune activation induces indoleamine 2,3-dioxygenase, which leads to lower plasma tryptophan and an increased synthesis of detrimental tryptophan catabolites (TRYCATs), both of which contribute to the onset of depression. Prog Neuropsychopharmacol Biol Psychiatry 2011, 35, 702-721, doi:10.1016/j.pnpbp.2010.12.017.
  7. Zhang, Q.; Shao, A.; Jiang, Z.; Tsai, H.; Liu, W. The exploration of mechanisms of comorbidity between migraine and depression. Journal of cellular and molecular medicine 2019, 23, 4505-4513, doi:10.1111/jcmm.14390.

Comment: There is a subsection in Results describing the correlation between 5-HIAA levels and PHQ-9.  Despite a small number of patients involved, it is worth, at least in the form of hypothesizing, to mention that there are two groups of two groups – one which has a strong negative correlation and another one which has a weak negative correlation of the parameters.

Answer: We have added the following sentence to the Results section:

“Moreover, two subgroups of migraine patients can be considered: one that has a strong negative correlation and another one that has a weak negative correlation between 5-HIAA levels and PHQ-9 scores.”

Comment: The Discussion section has a paragraph dedicated to this topic and starts with the sentence “Migraine is frequently associated with symptoms of anxiety and/or depression (reviewed in [39,40]).”. However, it is too general to understand how similar the presented results and data on 5-HIAA levels in depression are. This comparison is critical for understanding how specific can be 5-HIAA as a migraine biomarker.

Answer: Please see our answer to the last but one comment.

Results

Comment: 1) “2.1. Characteristics of Migraine Patients” section should have information on the control group as well as a demonstration of no statistical difference between the groups in comparable parameters e.g. age, sex etc.

Answer: The comparable parameters were age and sex only, because other characteristics were attributed to migraine. We have added the following sentence to the “Materials and methods” section:

“The control group was sex- and age-matched to migraine patients (Tables S1 and S2 in Supplementary Material). Occurrence of headaches and signs of depression or/and anxiety, except for very occasional cases with recognized causes, was an exclusion criterion for the control group.”

Comment: 2) Figures 3, 4, and 6 have legends misplaced.

Answer: We have tried to fix together figures with their legends.

Comment: 3) There is one likely outlier in MHD and MMD data sets. Was the test for outliers performed?

Answer: We have added the following fragment to the Results section:

“The graphs in Figures 4 and 5 may contain outliers, one per graph. These points can be identified as outliers with the ROUT (Prism) and the Grubbs’ test with moderate aggressiveness. However, we did not remove these results as our population was too small to conclude about the biological or medical significance of these results.”

Comment: 4) Fig 3-6 have to have results of control measurements presented as well. Visualization of variability in the control data set is important for understanding of the results. It can be a data point distribution at 0 of X-axes or a separate panel.

Answer: We are sorry, but we cannot follow this remark. Figs. 3-5 contain information on migraine severity, monthly migraine days, and monthly headache days; therefore, they should not be applied to the migraine-free control group. As per Fig. 6, we have added the following fragment to the “Materials and Methods” section:

“The occurrence of headaches and signs of depression or/and anxiety, except for very occasional cases with a recognized cause, was an exclusion criterion for the control group.”

Reviewer 3 Report (New Reviewer)

Comments and Suggestions for Authors

The investigators measure a range of substances in urine of individuals both with and without a history of migraine finding group- specific differences. They suggest these findings may have utility in diagnosis of migraine and as a measure of severity.

I think the manuscript would benefit by some more context initially, making clear that the search for biomarkers for migraine has been very extensive, of which the focus of the present manuscript is but one.  It would be useful, I think, to cite Demartini et al. Biomarkers of migraine: an integrated evaluation of preclinical and clinical findings (In J Mol Sciences 2023 24, 5334) which is very broad in scope but which also specifically encourages more research targeting tryptophan and its metabolites (I am not an author on that work).

At several points, the manuscript seems casually dismissive of clinical processes employed to make the diagnosis of migraine. Citing recent diagnostic criteria for migraine would be appropriate.

In addition, the investigators seem to imply that an objective (urine/blood test) result might serve to be a more valid indicator of migraine than clinical assessment (involving detailed history taking). Given that currently clinical diagnosis is the gold standard, there are clearly conceptual and practical additional challenges in showing that an alternative is better.

The existing research efforts to identify biomarkers for migraine have shown that variability exists according to gender, phase (ictal, interictal) together with other characteristics (e.g. with or without aura) etc. and those limitations to my mind warrant more attention when considering the possible value of 5-HIAA. If 5-HIAA were to be introduced as a diagnostically ‘marker of migraine occurrence and severity’ it would presumably need to aid in distinguishing migraine from non-migraine migraine mimicks. In order to validate its capacity to be both sensitive and specific, other studies which included conditions potentially confused with migraine would surely need to be undertaken. In addition, diagnostic properties for migraine subtypes etc. would need to be clarified. I think these sorts of considerations should also be noted.

Line 79, ‘To verify this hypothesis’ should be re written, substituting another word for verify (e.g. investigate or explore).

Line 275 ‘it is difficult to standardize criteria for migraine’ Why no citation of criteria or acknowledgement of efforts to standardize diagnosis?

The paragraph beginning line 320 seems odd. I don’t see why the lack of correspondence between questionnaire responses and 5-HIAA ‘underlies a need for more objective criteria for migraine characteristics’. Similarly, the following paragraph suggesting that 5-HIAA might be used to identify patients who might have psychiatric needs (i.e. privileging a pathology test over clinical inquiry around important symptoms) seems odd and very premature. 

I think the abstract should be substantially revised. The conclusion is not justified based on the results of this limited small study. Also, I don’t think the last paragraph (beginning line 377) is justified. Migraine is not the only cause of low 5-HIAA, not comparison of the 5-HIAA levels in conditions that might be confused with migraine have been done (or even discussed).

The limitations should be more comprehensively discussed.

Comments on the Quality of English Language

Adequate

Author Response

The investigators measure a range of substances in urine of individuals both with and without a history of migraine finding group- specific differences. They suggest these findings may have utility in diagnosis of migraine and as a measure of severity.

Comment: I think the manuscript would benefit by some more context initially, making clear that the search for biomarkers for migraine has been very extensive, of which the focus of the present manuscript is but one.  It would be useful, I think, to cite Demartini et al. Biomarkers of migraine: an integrated evaluation of preclinical and clinical findings (In J Mol Sciences 2023 24, 5334) which is very broad in scope but which also specifically encourages more research targeting tryptophan and its metabolites (I am not an author on that work).

Answer: We have added the following fragment to the Introduction section:

“The diagnosis of migraine is based on information provided by patients that can be biased, despite the precise criteria of The International Classification of Headache Disorders ICH-3 (R1). Therefore, there is a need to supplement the traditional diagnostic tools with more objective molecular markers. This does not mean that these markers are to replace the ICHD-3-based diagnosis, but they may be useful in resolving doubts about the traditional diagnosis. Despite the difficulties in obtaining the target material, there are intense works on molecular markers of migraine (reviewed in [4]).”

with new references:

  1. Demartini, C.; Francavilla, M.; Zanaboni, A.M.; Facchetti, S.; De Icco, R.; Martinelli, D.; Allena, M.; Greco, R.; Tassorelli, C. Biomarkers of Migraine: An Integrated Evaluation of Preclinical and Clinical Findings. International journal of molecular sciences 2023, 24, doi:10.3390/ijms24065334.

Comment: At several points, the manuscript seems casually dismissive of clinical processes employed to make the diagnosis of migraine. Citing recent diagnostic criteria for migraine would be appropriate.

Answer: We have added the following fragment to the Introduction section:

“The current ICHD-3, diagnostic criteria for migraine (migraine without aura, common migraine, hemicrania simplex) describe migraine as recurrent headache disorder manifesting in attacks lasting 4–72 hours [R1]. Typical characteristics of the headache are unilateral location, pulsating quality, moderate or severe intensity, aggravation by routine physical activity and association with nausea and/or photophobia and phonophobia. The diagnostic criteria are at least five attacks lasting 4–72 hours (when untreated or unsuccessfully treated) and headache has at least two of the following four characteristics: unilateral location, pulsating quality, moderate or severe pain intensity, aggravation by or causing avoidance of routine physical activity (e.g., walking or climbing stairs). During headache at least one of the following: nausea and/or vomiting photophobia and phonophobia. Not better accounted for by another ICHD-3 diagnosis. There are several notes and comments for these criteria, including that one or a few migraine attacks may be difficult to distinguish from symptomatic migraine-like attacks. Therefore, sometimes clinicians may encounter problems with distinguishing migraine from migraine-like headaches. IHCD-3 also provides criteria for other kinds of migraine.”

Comment: In addition, the investigators seem to imply that an objective (urine/blood test) result might serve to be a more valid indicator of migraine than clinical assessment (involving detailed history taking). Given that currently clinical diagnosis is the gold standard, there are clearly conceptual and practical additional challenges in showing that an alternative is better.

Answer: No, we did not intend to argue that any current molecular test might be more reliable in migraine diagnosis than the diagnosis based on the ICHD-3 criteria. We cannot find any statement in our original manuscript to support such (hypo)thesis. Moreover, we have added some text in its revised version to underline the differences between the clinical assessment and auxiliary molecular markers.

Comment: The existing research efforts to identify biomarkers for migraine have shown that variability exists according to gender, phase (ictal, interictal) together with other characteristics (e.g. with or without aura) etc. and those limitations to my mind warrant more attention when considering the possible value of 5-HIAA. If 5-HIAA were to be introduced as a diagnostically ‘marker of migraine occurrence and severity’ it would presumably need to aid in distinguishing migraine from non-migraine migraine mimicks. In order to validate its capacity to be both sensitive and specific, other studies which included conditions potentially confused with migraine would surely need to be undertaken. In addition, diagnostic properties for migraine subtypes etc. would need to be clarified. I think these sorts of considerations should also be noted.

Answer: Indeed! We have added the following text to the Discussion section:

“One of the main problems with the research efforts to identify biomarkers for migraine is that they may be sensitive not only to migraine and its characteristics but also to other factors, which, in turn, may influence migraine and sex and age are likely the most pronounced factors that could be taken into considerations. These parameters might be balanced as long as we compare migraine patients with the control individuals. Still, studies with migraine characteristics within migraine patients should have included other parameters as confounding factors. This is a limitation of our study, but the rationale for undertaking such analysis should be supported by further studies on the sensitivity of 5-HIAA to these possibly confounding factors and in migraine-like patients.”

Comment: Line 79, ‘To verify this hypothesis’ should be re written, substituting another word for verify (e.g. investigate or explore).

Answer: We have changed:

“To verify this hypothesis…”

into:

“To explore this hypothesis…”

Comment: Line 275 ‘it is difficult to standardize criteria for migraine’ Why no citation of criteria or acknowledgement of efforts to standardize diagnosis?

Answer: Because we have added text on the migraine criteria and the role of molecular markers in comparison with clinical assessment in response to the previous comments, we remove that sentence in the revised version of our manuscript.

Comment: The paragraph beginning line 320 seems odd. I don’t see why the lack of correspondence between questionnaire responses and 5-HIAA ‘underlies a need for more objective criteria for migraine characteristics’. Similarly, the following paragraph suggesting that 5-HIAA might be used to identify patients who might have psychiatric needs (i.e. privileging a pathology test over clinical inquiry around important symptoms) seems odd and very premature.

Answer: We are sorry that the lines 320-326 (in the original version) paragraph was carelessly written and in the light of the modifications to the text we introduced in response to previous comments, it is redundant. Similarly, we have removed the following fragment from the subsequent paragraph:

“Therefore, the result we obtained may be considered a small contribution to the link between migraine and mental disorders suggesting that TRP metabolism in general, and 5-HIAA in particular, may contribute to the mechanism underlying that association. However, it is not justified to widely discuss our results in a comparison with other studies performed with specialized methods.”

Comment: I think the abstract should be substantially revised. The conclusion is not justified based on the results of this limited small study. Also, I don’t think the last paragraph (beginning line 377) is justified. Migraine is not the only cause of low 5-HIAA, not comparison of the 5-HIAA levels in conditions that might be confused with migraine have been done (or even discussed).

Answer: The abstract has been rewritten. In particular, the last sentence:

“In conclusion, the urinary 5-HIAA level may be an easy-to-determine marker of migraine occurrence and severity and may contribute to the identification of migraine patients with depression.”

has been changed into:

“In conclusion, the urinary 5-HIAA level may be further explored to assess its suitability as an easy-to-determine marker of migraine.”

Furthermore, the last paragraph of the Discussion section has been changed from:

“In summary, the urinary concentration of 5-HIAA may be considered a marker of episodic migraine in the interictal period and may serve to identify these migraine patients who are at increased risk of migraine-associated mental disorders.”

into:

“In summary, the urinary concentration of 5-HIAA is negatively correlated with migraine occurrence and its characteristics and may be further explored to assess its usefulness as a non-invasive and easy-to-determine molecular marker in migraine.”

Comment: The limitations should be more comprehensively discussed.

Answer: We have added the following fragment to the paragraph discussing the limitations of our work:

“As stated above, our work did not include research on the dependence of the urinary 5-HIAA expression on other than migraine factors, including sex and age. Furthermore, 5-HIAA should be explored in different variants of migraine, e.g., migraine with aura, chronic migraine, especially in the context of the disease chronification, familial hemiplegic migraine as well as migraine-like symptoms.”

Round 2

Reviewer 3 Report (New Reviewer)

Comments and Suggestions for Authors

On line 109 the fragment 'To verify this hypothesis' is still there, despite the investigators agreeing with a recommended change. 

Author Response

Comment: On line 109 the fragment 'To verify this hypothesis' is still there, despite the investigators agreeing with a recommended change.

Answer: We are sorry that the recommended change escaped our attention; we have introduced it in the revised manuscript.

This manuscript is a resubmission of an earlier submission. The following is a list of the peer review reports and author responses from that submission.

Round 1

Reviewer 1 Report

Comments and Suggestions for Authors

The aim of this study was to investigate urinary tryptophan (TRP) metabolites and their correlation with markers of anxiety/depression in patients with low-frequency migraine. The data show that there is no significant difference between the TRP levels of patients and healthy subjects. However, migraine patients had lower levels of 5-Hydroxyindoleacetic acid and kynurenic acid than controls and higher levels of kynurenine and quinolinic acid. The 5-HIAA/TRP ratio was higher in the patient group than in the controls, suggesting an increase in the activity of indoleamine 2,3-dioxygenase (IDO), which is involved in the degradative metabolism of TRP. This change was associated with an increased KYN/TRP ratio in migraine patients. In contrast, the KYN/KYNA ratio in healthy patients was higher than in patients but not significantly so, suggesting that KYNA was metabolized in both groups. MMD, like MIDAS was negatively correlated with 5-HIAA levels and the 5-HIAA/TRP ratio. The number of headaches per month was correlated only with the urinary 5-HIAA levels in migraine patients. A negative correlation was also observed between PHQ-9 scores and urine concentration of 5-HIAA.

The topic of the manuscript is interesting, although not new. There are some points that could be better addressed.

-          The references used in the sentence "The KYN pathway of TRP metabolism is gaining an emerging role in migraine pathogenesis, supported by its involvement in the pathogenesis of functional gastrointestinal disorders and the functioning of the gut-brain-microbiota axis [6,10-12]" (lines 63-65) are not appropriate.

-          The authors mentioned the link to the microbiome. I wonder why they did not consider looking at this in the same patients.

-          The manuscript should include and discuss studies on metabolites of the kynurenine pathway in migraineurs.

-          A mechanism for 5-HIAA in migraine should also be proposed, as it is a possible biomarker for neurological and psychiatric diseases (doi: 10.15171/apb.2019.044). Some studies on urinary 5-HIAA in migraine have not been discussed. Previous research (doi: 10.1007/978-1-4615-4709-9; doi: 10.1046/j.1468-2982.1986. 0604205.x) found higher 5-HIAA levels in urine. In contrast, others observed lower amounts of the metabolite. These studies suggest that the major metabolite of 5‐HT, 5‐HIAA, may vary during periods of migraine.

-          Sample size calculation is not reported as well as primary and secondary outcomes.

-          When did the urine samples were collected in migraine patients? I guess, not during the ictal period, but how many days far from it and from medications?

-          Why the Authors did not also consider evaluating TRP metabolism in blood samples?

-          The procedure used to determine TRP metabolites by means of LC–MS/MS should be briefly reported or at least a reference must be indicated.

-          Table 1 is not very informative. I would suggest adding the means and standard deviations or frequencies with percentages, where applicable.

-          The methodology for psychological assessment should be explained in the Method section. Additionally, it is unclear how the presence of mood disorders was evaluated, as reported in Table 1. All of this needs to be explained, including the self-report questionnaires used.

-          Any significant differences in these aspects between the two groups of participants should be reported. I’d suggest to add a table comparing the two groups, with means and prevalence. It is also unclear why only the PHQ-9 is considered for analysis when the authors also report other instruments.

-          There are many typos across the paper. Please revise carefully.

-          Ensure that all legends for tables and figures are placed in the correct location.

-         The section of the discussion dedicated to psychological results should also be explained in relation to the literature. At present, it is merely a description of the findings.

Author Response

Referee #1

The aim of this study was to investigate urinary tryptophan (TRP) metabolites and their correlation with markers of anxiety/depression in patients with low-frequency migraine. The data show that there is no significant difference between the TRP levels of patients and healthy subjects. However, migraine patients had lower levels of 5-Hydroxyindoleacetic acid and kynurenic acid than controls and higher levels of kynurenine and quinolinic acid. The 5-HIAA/TRP ratio was higher in the patient group than in the controls, suggesting an increase in the activity of indoleamine 2,3-dioxygenase (IDO), which is involved in the degradative metabolism of TRP. This change was associated with an increased KYN/TRP ratio in migraine patients. In contrast, the KYN/KYNA ratio in healthy patients was higher than in patients but not significantly so, suggesting that KYNA was metabolized in both groups. MMD, like MIDAS was negatively correlated with 5-HIAA levels and the 5-HIAA/TRP ratio. The number of headaches per month was correlated only with the urinary 5-HIAA levels in migraine patients. A negative correlation was also observed between PHQ-9 scores and urine concentration of 5-HIAA.

The topic of the manuscript is interesting, although not new. There are some points that could be better addressed.

Comment:  The references used in the sentence "The KYN pathway of TRP metabolism is gaining an emerging role in migraine pathogenesis, supported by its involvement in the pathogenesis of functional gastrointestinal disorders and the functioning of the gut-brain-microbiota axis [6,10-12]" (lines 63-65) are not appropriate.

Answer: We are sorry, but we do not understand. Each of these references 6, 10-12 in the original submission is on either tryptophan metabolism with a special emphasis on its kynurenine pathway, or migraine and the gut-brain axis. Therefore, we have not changed the text in that sentence.

Comment: The authors mentioned the link to the microbiome. I wonder why they did not consider looking at this in the same patients.

Answer: We have added the following sentence to the 3. Discussion section:

“In our previous works, we considered tryptophan metabolism in the context of the gut-brain-microbiota axis [42-45]. However, those studies were performed on hospital patients, but this study enrolled patients of an outpatient clinic and most of them were not willing to donate blood or undergo the examination of the gut and the microbiota.”

with the references:

  1. Chojnacki, C.; Gąsiorowska, A.; Popławski, T.; Błońska, A.; Konrad, P.; Zajdler, R.; Chojnacki, J.; Blasiak, J. Reduced Intake of Dietary Tryptophan Improves Beneficial Action of Budesonide in Patients with Lymphocytic Colitis and Mood Disorders. Nutrients 2023, 15, doi:10.3390/nu15071674.
  2. Chojnacki, C.; Popławski, T.; Chojnacki, J.; Fila, M.; Konrad, P.; Blasiak, J. Tryptophan Intake and Metabolism in Older Adults with Mood Disorders. Nutrients 2020, 12, doi:10.3390/nu12103183.
  3. Chojnacki, C.; Popławski, T.; Konrad, P.; Fila, M.; Błasiak, J.; Chojnacki, J. Antimicrobial treatment improves tryptophan metabolism and mood of patients with small intestinal bacterial overgrowth. Nutr Metab (Lond) 2022, 19, 66, doi:10.1186/s12986-022-00700-5.
  4. Chojnacki, C.; Popławski, T.; Konrad, P.; Fila, M.; Chojnacki, J.; Błasiak, J. Serotonin Pathway of Tryptophan Metabolism in Small Intestinal Bacterial Overgrowth-A Pilot Study with Patients Diagnosed with Lactulose Hydrogen Breath Test and Treated with Rifaximin. Journal of clinical medicine 2021, 10, doi:10.3390/jcm10102065.

Comment: The manuscript should include and discuss studies on metabolites of the kynurenine pathway in migraineurs.

Answer: This is an experimental work, and we must keep the right proportion between its sections. In the original submission, a discussion on 5-HIAA has almost 1 page and we cannot add 2-3 pages presenting results on migraine and components of the kynurenine pathway, the more that the compounds of the kynurenine pathway of TRP metabolism we studied were not changed.

Comment: A mechanism for 5-HIAA in migraine should also be proposed, as it is a possible biomarker for neurological and psychiatric diseases (doi: 10.15171/apb.2019.044). Some studies on urinary 5-HIAA in migraine have not been discussed. Previous research (doi: 10.1007/978-1-4615-4709-9; doi: 10.1046/j.1468-2982.1986. 0604205.x) found higher 5-HIAA levels in urine. In contrast, others observed lower amounts of the metabolite. These studies suggest that the major metabolite of 5‐HT, 5‐HIAA, may vary during periods of migraine.

Answer: The doi: 10.1046/j.1468-2982.1986. 0604205.x paper has been cited and discussed in our original submission as the reference [36].

We have added the following fragment in the Discussion section:

“Urinary level of 5-HIAA was observed not to change in young men but it decreased in female migraine patients when compared with their sex-matched controls [38]. No association was observed between 5-HIAA excretion and the characteristics of migraine.”

with

P doi: 10.1007/978-1-4615-4709-9 (Bousser, M.G.; Elghozi, J.L.; Laude, D.; Soisson, T. Urinary 5-HIAA in migraine: evidence of lowered excretion in young adult females. Cephalalgia : an international journal of headache 1986, 6, 205-209, doi:10.1046/j.1468-2982.1986.0604205.x.)

Comment: Sample size calculation is not reported as well as primary and secondary outcomes.

Answer: Our study is not a clinical trial. We recruited as many patients as we could in the time devoted to the study. Our primary and secondary goals are presented in the last two paragraphs of the 1. Introduction section.

Comment: When did the urine samples were collected in migraine patients? I guess, not during the ictal period, but how many days far from it and from medications?

Answer: We have added the following sentence in the 2. Material and Methods section:

“The urine samples were taken in the interictal period 10-30 days after the last migraine attack, which corresponded to the last administration of headache abortive drugs.”

Comment: Why the Authors did not also consider evaluating TRP metabolism in blood samples?

Answer: Please see our answer to the question about the microbiome.

Comment: The procedure used to determine TRP metabolites by means of LC–MS/MS should be briefly reported or at least a reference must be indicated.

Answer: We have changed the sentence:

“L-tryptophan and its following metabolites: 5-HIAA, KYN, KYNA, and QA were determined in urine with liquid chromatography–tandem mass spectrometry (LC–MS/MS—Ganzimmun Diagnostics AG, Mainz, Germany; D-ML-13147–01-01).”

into:

“L-tryptophan and its following metabolites: 5-HIAA, KYN, KYNA, and QA were determined in urine with liquid chromatography–tandem mass spectrometry (LC–MS/MS—Ganzimmun Diagnostics AG, Mainz, Germany; D-ML-13147–01-01) as described elsewhere [48] with some modifications [49].”

with new references:

  1. Zhu, W.; Stevens, A.P.; Dettmer, K.; Gottfried, E.; Hoves, S.; Kreutz, M.; Holler, E.; Canelas, A.B.; Kema, I.; Oefner, P.J. Quantitative profiling of tryptophan metabolites in serum, urine, and cell culture supernatants by liquid chromatography-tandem mass spectrometry. Anal Bioanal Chem 2011, 401, 3249-3261, doi:10.1007/s00216-011-5436-y.
  2. Chung, S.H.; Yoo, D.; Ahn, T.B.; Lee, W.; Hong, J. Profiling Analysis of Tryptophan Metabolites in the Urine of Patients with Parkinson's Disease Using LC-MS/MS. Pharmaceuticals (Basel) 2023, 16, doi:10.3390/ph16101495.

Comment: Table 1 is not very informative. I would suggest adding the means and standard deviations or frequencies with percentages, where applicable.

Answer: We have replaced Table 1 from the original submission with a table containing the mean, standard deviation, and min-max interval, where possible. However, for certain quantities, e.g., mood disorders, aura, and treatment, we indicated only the number of affected patients. Moreover, we have added information about sex and frequency of attacks per quarter.

Comment: The methodology for psychological assessment should be explained in the Method section. Additionally, it is unclear how the presence of mood disorders was evaluated, as reported in Table 1. All of this needs to be explained, including the self-report questionnaires used.

Answer: We have added the following sentence in 4. Material and Methods

“Mood disorder was assessed with a non-standard questionnaire based on the self-reported presence of 5 out of the 8 symptoms of sad mood, - hipo- or insomnia, feelings of guilt, decreased energy levels, decreased concentration, decreased appetite, decrease in pleasurable activities (anhedonia), and increased or decreased psychomotor activity [48]. Also, GAD-7, Generalized Anxiety Disorder 7-item and PHQ-9, Patient Health Questionnaire 9 were used to assess patients’ mental health.”

with the new reference:

  1. Sekhon, S.; Gupta, V. Mood Disorder. In StatPearls; StatPearls Publishing Copyright © 2024, StatPearls Publishing LLC.: Treasure Island (FL), 2024.

Comment:  Any significant differences in these aspects between the two groups of participants should be reported. I’d suggest to add a table comparing the two groups, with means and prevalence. It is also unclear why only the PHQ-9 is considered for analysis when the authors also report other instruments.

Answer: We are sorry, but we do not understand. We studied psychological characteristics only in migraine patient group in possible correlation with TRP metabolites and their ratios. We did not compare any two groups in this regard, so we cannot add such a table. Only PHQ-9 is presented because it was the only characteristic that was significantly correlated with 5-HIAA level.

Comment: There are many typos across the paper. Please revise carefully.

Answer: We are sorry, but we cannot see them.

Comment: Ensure that all legends for tables and figures are placed in the correct location.

Answer: We have adjusted the position of the table titles and figure legends, but we cannot guarantee that they will be kept in an MS Word file on every computer.

Comment: The section of the discussion dedicated to psychological results should also be explained in relation to the literature. At present, it is merely a description of the findings.

Answer: We are sorry, but we are afraid that we have nothing more to say on this subject as we employed simple, standard tests to assess the mental state of the patients and their results are just a small contribution that justifies further research.

Reviewer 2 Report

Comments and Suggestions for Authors

The submitted manuscript presents interesting data on the status of kynurenine pathway in patients with migraines. The design of the study was appropriate and, despite relatively low number of enrolled individuals, the results bring additional input into our understanding of disease pathogenesis.

However, I would like to ask the authors to include the following aspects  in the manuscript

1) Please, add the information whether samples were collected during interictal period or during an attack of migraine in Materials and Methods and Abstract.

2) As the authors analyzed the status of kynurenine pathway in migraine, the discussion must include the comparison of their data, in terms of specific metabolites, with other results. At present, the discussion provides comprehensive analysis of the data on serotonin but not on KYNA, KYN or QUIN.

3) Could the authors, please, explain why the 3-hydroxykynurenine levels were not measured and add it to the discussion as limitation?

4) Please, add to the Discussion  information that an increased conversion of TRP to KYN with a decrease of KYNA/QUIN ratio implies that there may be also a change in the level of 3-hydroxykynurenine.

5) Please, include in the manuscript the following references:

J Headache Pain. 2021 Jun 25;22(1):60. doi: 10.1186/s10194-021-01239-1. PMID: 34171996; PMCID: PMC8229298;

J Headache Pain. 2015;17(1):27. doi: 10.1186/s10194-016-0620-2. Epub 2016 Mar 22. PMID: 27000870; PMCID: PMC4801826.

Author Response

Referee #2

The submitted manuscript presents interesting data on the status of kynurenine pathway in patients with migraines. The design of the study was appropriate and, despite relatively low number of enrolled individuals, the results bring additional input into our understanding of disease pathogenesis.

However, I would like to ask the authors to include the following aspects  in the manuscript

Comment: 1) Please, add the information whether samples were collected during interictal period or during an attack of migraine in Materials and Methods and Abstract.

Answer: We have added the following sentence in the 2. Material and Methods section:

“The urine samples were taken in the interictal period 10-30 days after the last migraine attack, which corresponded to the last administration of headache abortive drugs.”

Comment: 2) As the authors analyzed the status of kynurenine pathway in migraine, the discussion must include the comparison of their data, in terms of specific metabolites, with other results. At present, the discussion provides comprehensive analysis of the data on serotonin but not on KYNA, KYN or QUIN.

Answer: We are sorry, but we cannot extend too much our discussion section by elements that are not directly related to the positive results we obtained as it would be inadequate to the results. We have added the following sentence to 3. Discussion:

“Although we did not observe changes in the KYN, KYNA, and QA levels, we cannot exclude alterations in the kynurenine pathway of TRP metabolism in our patients, as we did not measure the level of all metabolites of that pathway. In particular, 3-hydroxykynurenine levels were not measured. More evidence on the role of the KYN pathway of TRP metabolism in migraine and headaches can be found in other works, e.g., [48-51].”

with new references:

  1. Curto, M.; Lionetto, L.; Negro, A.; Capi, M.; Perugino, F.; Fazio, F.; Giamberardino, M.A.; Simmaco, M.; Nicoletti, F.; Martelletti, P. Altered serum levels of kynurenine metabolites in patients affected by cluster headache. J Headache Pain 2015, 17, 27, doi:10.1186/s10194-016-0620-2.
  2. Fila, M.; Chojnacki, C.; Chojnacki, J.; Blasiak, J. The kynurenine pathway of tryptophan metabolism in abdominal migraine in children - A therapeutic potential? Eur J Paediatr Neurol 2024, 48, 1-12, doi:10.1016/j.ejpn.2023.11.001.
  3. Fila, M.; Chojnacki, J.; Pawlowska, E.; Szczepanska, J.; Chojnacki, C.; Blasiak, J. Kynurenine Pathway of Tryptophan Metabolism in Migraine and Functional Gastrointestinal Disorders. International journal of molecular sciences 2021, 22, doi:10.3390/ijms221810134.
  4. Tuka, B.; Nyári, A.; Cseh, E.K.; Körtési, T.; Veréb, D.; Tömösi, F.; Kecskeméti, G.; Janáky, T.; Tajti, J.; Vécsei, L. Clinical relevance of depressed kynurenine pathway in episodic migraine patients: potential prognostic markers in the peripheral plasma during the interictal period. J Headache Pain 2021, 22, 60, doi:10.1186/s10194-021-01239-1.

Comment: 3) Could the authors, please, explain why the 3-hydroxykynurenine levels were not measured and add it to the discussion as limitation?

Answer: Please see our answer to the previous comment. We have also added the following statement to 3. Discussion:

“Another limitation of our study is not to measure the level of 3-hydroxykynurenine, a product of the KYN pathway of TRP metabolism, especially since this molecule is reported to exert a dual, pro- or anti-oxidative action in the central nervous system  [47].”

with the new reference

  1. Colín-González, A.L.; Maldonado, P.D.; Santamaría, A. 3-Hydroxykynurenine: An intriguing molecule exerting dual actions in the Central Nervous System. NeuroToxicology 2013, 34, 189-204, doi:https://doi.org/10.1016/j.neuro.2012.11.007.

Comment: 4) Please, add to the Discussion  information that an increased conversion of TRP to KYN with a decrease of KYNA/QUIN ratio implies that there may be also a change in the level of 3-hydroxykynurenine.

Answer: We have added the following sentence to continue our statement on the limitations:

“Moreover, increased conversion of TRP to KYN with a decrease of KYNA/QA ratio suggests that the level of 3-hydroxykynurenine might change [48].

with the new reference:

  1. Fathi, M.; Vakili, K.; Yaghoobpoor, S.; Tavasol, A.; Jazi, K.; Hajibeygi, R.; Shool, S.; Sodeifian, F.; Klegeris, A.; McElhinney, A.; et al. Dynamic changes in metabolites of the kynurenine pathway in Alzheimer's disease, Parkinson's disease, and Huntington's disease: A systematic Review and meta-analysis. Front Immunol 2022, 13, 997240, doi:10.3389/fimmu.2022.997240.

Comment: 5) Please, include in the manuscript the following references:

J Headache Pain. 2021 Jun 25;22(1):60. doi: 10.1186/s10194-021-01239-1. PMID: 34171996; PMCID: PMC8229298;

J Headache Pain. 2015;17(1):27. doi: 10.1186/s10194-016-0620-2. Epub 2016 Mar 22. PMID: 27000870; PMCID: PMC4801826.

Answer: Please see our answer to the 2) comment.

Round 2

Reviewer 1 Report

Comments and Suggestions for Authors

The authors have tried to improve the manuscript in some places. However, they have failed to discuss the results of their study, which are no different from previously published results. It is well known that monitoring serum Try levels and urinary 5-HIAA concentrations may not only indirectly represent serotonin levels but also have implications for early diagnosis and monitoring of depression.  The calculation for the sample size must be conducted regardless of whether it is a clinical trial.

Author Response

Comment: he authors have tried to improve the manuscript in some places. However, they have failed to discuss the results of their study, which are no different from previously published results. It is well known that monitoring serum Try levels and urinary 5-HIAA concentrations may not only indirectly represent serotonin levels but also have implications for early diagnosis and monitoring of depression.  

Answer: This is a migraine subjected manuscript. We have not enough deep results to lead psychological discussion.

Comment: The calculation for the sample size must be conducted regardless of whether it is a clinical trial.

Answer: As we pointed in the answers to the round 1 comments, we did not calculate sample size.

Reviewer 2 Report

Comments and Suggestions for Authors

The manuscript is now suitable for publication

Author Response

Comment: The manuscript is now suitable for publication

Answer: Thank you!

Round 3

Reviewer 1 Report

Comments and Suggestions for Authors

The authors refuse to comply with the requests.